# Calcination Enhances the Aflatoxin and Zearalenone Binding Efficiency of a Tunisian Clay

**DOI:** 10.3390/toxins11100602

**Published:** 2019-10-16

**Authors:** Roua Rejeb, Gunther Antonissen, Marthe De Boevre, Christ’l Detavernier, Mario Van de Velde, Sarah De Saeger, Richard Ducatelle, Madiha Hadj Ayed, Achraf Ghorbal

**Affiliations:** 1Université de Sousse, Institut Supérieur Agronomique de Chott-Mariem, LR18AG01, ISA-CM-BP, 47, Sousse 4042, Tunisia; mediha.ayed@yahoo.fr; 2Department of Pathology, Bacteriology and Avian Diseases, Faculty of veterinary medicine, Ghent University, Salisburylaan 133, 9820 Merelbeke, Belgium; Gunther.Antonissen@UGent.be (G.A.); Richard.Ducatelle@UGent.be (R.D.); 3Department of Pharmacology, Toxicology and Biochemistry, Faculty of Veterinary Medicine, Ghent University, Salisburylaan 133, 9820 Merelbeke, Belgium; 4Department of Bioanalysis, Centre of Excellence in Mycotoxicology and Public Health, Faculty of Pharmaceutical Sciences, Ghent University, Ottergemsesteenweg 460, 9000 Ghent, Belgium; Marthe.DeBoevre@UGent.be (M.D.B.); Christel.Detavernier@UGent.be (C.D.); Mario.VandeVelde@UGent.be (M.V.d.V.);; 5Research Laboratory LR18ES33, National Engineering School of Gabes, University of Gabes, Avenue Omar Ibn El Khattab, Gabes 6029, Tunisia; achraf.ghorbal.issat@gmail.com

**Keywords:** aflatoxins, zearalenone, clay, purified, calcined, adsorption, pH

## Abstract

Clays are known to have promising adsorbing characteristics, and are used as feed additives to overcome the negative effects of mycotoxicosis in livestock farming. Modification of clay minerals by heat treatment, also called calcination, can alter their adsorption characteristics. Little information, however, is available on the effect of calcination with respect to mycotoxin binding. The purpose of this study was to characterize a Tunisian clay before and after calcination (at 550 °C), and to investigate the effectiveness of the thermal treatment of this clay on its aflatoxin B1 (AFB1), G1 (AFG1), B2 (AFB2), G2 (AFG2), and zearalenone (ZEN) adsorption capacity. Firstly, the purified clay (CP) and calcined clay (CC) were characterized with X-ray Fluorescence (XRF), X-ray Diffraction (XRD), Fourier transform infrared spectroscopy (FTIR-IR), cation exchange capacity (CEC), specific surface area (S_BET_), and point of zero charge (pH_PZC_) measurements. Secondly, an in vitro model that simulated the pH conditions of the monogastric gastrointestinal tract was used to evaluate the binding efficiency of the tested clays when artificially mixed with aflatoxins and zearalenone. The tested clay consisted mainly of smectite and illite. Purified and calcined clay had similar chemical compositions. After heat treatment, however, some changes in the mineralogical and textural properties were observed. The calcination decreased the cation exchange capacity and the specific surface, whereas the pore size was increased. Both purified and calcined clay had a binding efficacy of over 90% for AFB1 under simulated poultry GI tract conditions. Heat treatment of the clay increased the adsorption of AFB2, AFG1, and AFG2 related to the increase in pore size of the clay by the calcination process. ZEN adsorption also increased by calcination, albeit to a more stable level at pH 3 rather than at pH 7. In conclusion, calcination of clay minerals enhanced the adsorption of aflatoxins and mostly of AFG1 and AFG2 at neutral pH of the gastrointestinal tract, and thus are associated with protection against the toxic effects of aflatoxins.

## 1. Introduction

Mycotoxins are secondary metabolites produced by toxigenic fungi growing on a wide range of agricultural products [1]. The Food and Agriculture Organization (FAO) of the United Nations stated that nearly 25% of the cereal products is contaminated with mycotoxins [2].

Aflatoxins (AFs) are considered as the most important mycotoxins in food and feed because of their carcinogenicity (IARC monograph class I) and their high prevalence, especially in Southern regions [3,4]. AFs are a group of heterocyclic metabolites that are mainly produced by members of the genus *Aspergillus*, contaminating agricultural commodities. *Aspergillus* fungi are both found in the field as storage pathogens, therefore the AFs’ content is omnipresent, but with a tendency to increase during storage [5,6]. AFs cause serious health problems, economic losses, and deleterious effects on performance in a variety of farm animals including pigs, poultry, and cattle [7]. More specifically in poultry, AFs reduce the growth rate, decrease egg production, induce changes in organ weight, and increase the risk of disease [4]. In young chicks, AFs reduce immune competence and cause liver damage [8,9,10]. Aflatoxin B1 (AFB1) is the most potent of the aflatoxins, followed by AFG1, AFB2, and AFG2 [11]. AFB1 comprise a greater potency associated with the cyclopentenone ring of the B series, when compared with the six-membered lactone ring of the G series [12].

Zearalenone (ZEN) is a phenolic resorcyclic acid lactone mycotoxin produced by *Fusarium* fungi growing on cereal grains and derived products worldwide. *Fusarium* mycotoxins are primarily produced before harvest, on the field. ZEN is a potent estrogenic metabolite as it has the ability to bind to estrogen receptors and induces estrogenic alterations including uterine enlargement, swelling of the vulva and mammary glands, and pseudopregnancy [13,14].

To offset the negative effects of mycotoxins on animal health, a wide range of mycotoxin decontamination strategies has been reported in the literature [15,16,17]. Adding mycotoxin binders to the feed is probably the most common post-harvest mitigation approach [18,19,20]. Binders decrease the absorption of mycotoxins from the gastrointestinal (GI) tract into the blood circulation and target organs by adsorbing them on their surface [21], forming a binder-toxin complex which is eliminated through the fecal material [22]. Inclusion of mycotoxin binders improves the average daily gain and the average daily feed intake in pigs and reduced the injurious effect of AFs on body weight, feed conversion ratio (FCR), serum alanine aminotransferase (ALT), and urea concentration in broiler chickens [8,23]. However, it should be noted that adding a high dose of clay in the feed might cause nutrient deficiency by adsorbing micronutrients, vitamins and organic compounds, while also having negative effects on the bioavailability of minerals and trace elements [15,24]. In addition, the risk of the contamination of raw clays with metals and dioxins has to be considered [25]. Among mycotoxin binders, clay minerals are the largest group. Several aluminosilicate clays such as hydrated sodium calcium aluminosilicate (HSCAS), bentonite, montmorillonite, smectite, and zeolite have good binding efficiency to mycotoxins [26,27]. These clays mostly bind small mycotoxins, such as AFs and ochratoxin A, but have less binding affinity for the larger molecules of certain *Fusarium* toxins. ZEN can be adsorbed by only a limited number of binders with a large variation in binding capacity [28]. Effectiveness of binding also depends on the type, and the dosage of the binders. A Brazilian study showed that about 64% of the products on the market were ineffective in binding AFs [29,30].

Clays are used in their natural state or treated through various processes such as calcination, acid activation, pillaring, organic modification with polymers, or cation and anion exchange. These modified clays can be more effective in binding some mycotoxins than the untreated clay [31,32,33,34]. Calcination is a process in which clay minerals are heated to different temperatures [33]. It removes the water located in intra-crystalline tunnels, which changes the pore structure and surface proprieties [35,36]. Consequently, heat treatment has an impact on the specific surface area of clay, which is responsible for the adsorption capacity [37]. Calcined clays have been used in numerous studies as adsorbents to eliminate heavy metals and cationic dyes [38,39].

To the best of the authors’ knowledge, only one study investigated the effect of calcination on mycotoxin binding [40]. In this paper it was reported that the adsorption of AFB1 was reduced after calcination of a bentonite clay. It is questionable whether this observation can be extrapolated to other types of clay and other mycotoxins. Therefore, the purpose of the present study was to analyze the physical characteristics and to evaluate the binding of different mycotoxins before and after calcination of a montmorillonite type of clay from Tunisia.

## 2. Results

### 2.1. Chemical Characterization

XRF analyses revealed that purified native clay (CP) and its calcined form (CC) were mainly composed of silicon dioxide (SiO2), aluminum oxide (Al_2_O_3_), calcium oxide (CaO) and iron oxide (Fe_2_O_3_) (Table 1). In addition, the occurrence of magnesium oxide in both samples can be due to the presence of smectite, and also to a small amount of dolomite, which is confirmed by the results of XRD and FTIR-IR [41].

### 2.2. Infrared Spectroscopy Characterization (FTIR-ATR)

The FTIR patterns of the CP (Figure 1) exhibits several characteristic bands corresponding to the stretching vibrations of the surface hydroxyl groups (Si–Si–OH, or Al–Al–OH) at 3695 cm^−1^ which indicates the presence of kaolinite [40,41,42,43,44]. The spectrum shows a characteristic band of montmorillonite at 3620 cm^−1^ [43]. The band at 1432 cm^−1^ corresponds to carbonate [calcite (Ca CO_3_) or dolomite (Ca, Mg (CO_3_)_2_)] [44,45,46]. The band at 711 cm^−1^ corresponds to calcite [44,45]. The band at 518 cm^−1^ is due to Si–O–Al (octahedral) bending vibration [46,47]. Vibration at 1635 cm^−1^ was attributed to the bending of adsorbed water. The band of deformation near to 873 cm^−1^ indicates that the clay is octahedral [48], while the band near 910 cm^−1^ corresponds to an Al-O-H deformation characteristic of dioctahedral smectite [49].

After calcination, the water OH-bending (1635 cm^−1^) mode totally disappeared. This is a consequence of dehydroxylation and dehydration by the thermal treatment.

### 2.3. X-Ray Diffraction (XRD)

XRD characterization as presented in Figure 2 showed that CP and CC consisted mainly of calcic smectite, as demonstrated by the main peak near 14.10 Å and the additional peaks at 4.48 Å and 2.56 Å [50]. The XRD patterns confirmed the presence of kaolinite by the basal spacing at 7.16 Å, 3.84 Å, and 3.57 Å [45,50,51]. The characteristic reflection of dolomite was observed at 2.89 Å [45], while that of calcite was observed at 3.03 Å and 1.90 Å [45,50,52]. Both clays were also characterized by the presence of quartz, indicated by several peaks at 4.25 Å, 3.34 Å, 2.28 Å, 2.09 Å, and 1.87 Å [47,48,53,54]. Following calcination, the characteristic peaks of kaolinite at 7.16 Å and 3.57 Å disappeared, while a new peak appeared at 9.91 Å.

### 2.4. Cation Exchange Capacity (CEC)

The thermal treatment resulted in a decreased CEC (9.28 Cmol_(+)_ (kg^−1^) for CC) and a reduction in Ca, Mg and Na after calcination (Table 2).

### 2.5. BET Surface Analysis

Brauner-Emmett-Teller (BET) N_2_ adsorption/desorption analysis showed a surface area of CP 64.06 m^2^/g. Thermal treatment resulted in a decreased surface area: 44.42 m^2^/g for CC (Table 3) and an increase in pore size from 57.03 Å for CP to 66.72 Å for CC.

### 2.6. Point of Zero Charge (PZC)

As can be seen in Figure 3, the pH_PZC_ of CP was 9.94, while it was 10.03 for the CC.

### 2.7. Mycotoxin Binding Efficiency

At pH of 3, CC and CP were able to bind 100% of AFB1. However, compared to CP, CC had a higher (*p* < 0.05) adsorption capacity for AFB2, AFG1 and AFG2 (Table 4). Moreover, CP was not able to bind ZEN at pH 3, while the calcinated clay did adsorb ZEN (*p* < 0.05). Table 5 contains the results of mycotoxin adsorption of the CP and CC at pH 7. The adsorption of AFB1, AFB2, AFG1, and AFG2 at pH 7 was significantly higher (*p* < 0.05) for CC compared to CP. As almost no ZEN was bound to CP, the adsorption of ZEN at pH 7 was significantly higher (*p* < 0.05) for the CC (41 ± 12%) than the CP (1 ± 1%).

Based on these in vitro adsorption data, the in vitro binding efficiency of AFs and ZEN of the purified and calcined clay was predicted (Figure 4). The results show that both clays exhibited a strong in vitro binding affinity to AFs. Heat treatment of the clay improved the binding efficiency of ZEN (29 ± 18%) compared to the CP (0 ± 0%). The binding efficiency of AFB1 was 94 ± 3% and 99 ± 0% for the CP and CC, respectively. The effectiveness of CP and CC in binding AFB2 was 86 ± 9% and 99 ± 1%, respectively. The in vitro binding efficiency of AFG1 and AFG2 was significantly higher (*p* < 0.05) for the CC than the CP. AFG1 binding efficiency was 59 ± 18% for CP and 98 ± 2% for CC (*p* < 0.05). Furthermore, the binding efficiency of AFG2 was 15 ± 38% for the CP and 96 ± 2% for the CC. These results demonstrate that the thermal treatment of the clay enhances the binding efficiency of AFG1 and AFG2.

## 3. Discussion

Several reports have shown that adding phyllosilicate clay supplements to animal feed represents one of the most powerful prevention strategies for aflatoxicosis in livestock [21,55,56,57]. Heat treatment may alter the adsorbing properties of the clays, however, this has not been investigated much with respect to mycotoxins so far.

Our results show that the chemical composition of the studied clay was not affected after heat treatment. Similar results were found in another study with a different type of clay [40]. Indeed, there is no change in bentonite composition before and after calcination. The chemical composition of the studied clay is a typical composition of south-eastern Tunisian clays [45]. CP and CC were also characterized by a high calcium oxide content and a low percentage of sodium oxide which are in accordance with previous work [48]. 

Disappearance of the characteristic peaks of kaolinite following calcination, as observed by XRD, is probably due to the dehydroxylation of the kaolinite structure, leading to the collapse of clay mineral structure [40,50,58]. Aluminosilicate minerals are characterized by a CEC varying from 10 (meq/100 g) for kaolinite minerals to 100 (meq/100 g) for illite and smectite minerals [59]. The tested clay showed a low value of CEC which can be due to the poor crystallinity of smectite and to the presence of kaolinite, illite, and some impurities (calcite, quartz, dolomite, etc.). The decrease in CEC is due to the dehydration and dehydroxylation of smectite which results in a collapse of the interlayer [53,60]. This result is in agreement with previous reports [34,61] stating that the thermal treatment reduces the cation exchange capacity of clays. The surface area is an important property for clay adsorption capacity. Previous work reported that the surface area decreases with increasing temperatures of thermal treatment [33,46,54]. Calcination of clay at 560 °C causes an agglomeration of the particles leading to a 15% decrease in the surface area and an increase in the pore size [62]. According to the literature [35], the increase in the diameter of pores is due to a reduction of the basal spacing through dehydration of the interlayer spaces which is a consequence of a first collapse of smectite layers. Similar research observed that calcination of clay affects the interlayer space [63,64]. The pH_PZC_ is an important characteristic of minerals and usually used to define the state of the surface of a dispersed solid phase at the solid-electrolyte solution surface and may indicate the ionization of functional groups and their possible interaction with the mycotoxin molecules [65]. In addition, low-charge montmorillonites or other low-charge smectites would be more effective feed additives than higher-charge clays [34]. The surface is negative at pH>pH_PZC_, positive at pH<pH_PZC_ and neutral at pH=pH_PZC_ [66]. The obtained results for pH_PZC_ in our study indicated that the surface of the studied clays is positive at pH3 and pH7 and are moderately higher than the pH_PZC_ reported in the literature for Tunisian smectitec clays (8.2) [67]. The use of adsorbents mixed with food and feed is one of the prominent post-harvest approaches to protect against mycotoxins toxicity, which are supposed to bind efficiently mycotoxins in the gastrointestinal tract. In the current study, we evaluate the binding capacity of AFs and ZEN of purified and calcined clay. The in vitro ZEN and AFs adsorption is assessed at pH 3 and pH 7 which are representative for the GI tract of the monogastric animals. Similar models were successfully applied in previous in vitro experiments [21,59,68]. Regarding adsorption and desorption of ZEN, the adsorption of ZEN by clay is usually lower than aflatoxins [21]. This could be because of the low polarity of ZEN compared to AFs [19]. Furthermore, ZEN has a more spherical molecular geometry than the planar structure of AFs [18]. Moreover, in a recent study [21], it was reported that ZEN is significantly less adsorbed in alkaline than in acid and neutral conditions. De Mil et al. [18] suggested that the pH may influence the phenolic hydroxyl group of ZEN or the ionization-state of the functional groups of the mycotoxin binders, and thereby alter the adsorption by the ionic interactions.

Our findings confirmed a better adsorption capacity for ZEN for the calcined clay than the purified native clay. It has been reported that modified clays have been developed to improve ZEN adsorption, which provide sufficient space between the layers to react with mycotoxin with a relatively less polarity with the appropriate electrical charging [69,70,71,72]. These modified surface properties lead to greater hydrophobicity by exchanging the structural load balance cations with high molecular weight quaternary amines [27]. Feng et al. [73] have concluded that the modified clays have led to low desorption rates, with higher ZEN adsorption than the non-modified clay. The increase in adsorption capacity of AFs and ZEN with calcined clay in our study might be related to the increase in pore size (Table 3), and the decrease in CEC (Table 2) of clay after heat treatment at 550 °C. Chen et al. [35] reported that calcination of palygorskite at high temperatures improves dye adsorption efficiency, which is basically associated with a larger size and a wider size distribution of pores. In addition to the pore size, CEC play an important role in the adsorption phenomenon [18]. Exchangeable cations neutralize the interlayer charges in phyllosilicates, and are involved in the binding of AFB1 [7,74]. Deng et al. [75] suggested that layer charge density and the type of exchange cations have a distinct influence on the adsorption of AF. Recent research [43] demonstrated that heating of bentonite improves the clays’ adsorption affinity and capacity of AFB1, which is mainly because of the reduction of cation exchange capacity. Furthermore, AFB1 is a polar mycotoxin and contains ß-carbonyl, which is involved in the adsorption process [55]. According to Prapapanpong et al. [21], the adsorption process involves the exchange of electrons of the metallic cation on the surface of the adsorbent, especially the positive charge of calcium ions on each layer of clay. A hypothesis proposed by Jaynes et al. [76] discussed the possibility that aflatoxins can be captured at multiple locations on hydrated sodium calcium aluminosilicate (HSCAS) surfaces, as well as between HSCAS inter-layers.

## 4. Conclusions

In this study, calcination improved the adsorption efficacy of the clay for AFB2, AFG1, AFG2, and ZEN. The adsorption-desorption rate showed that the pH condition in the GI tract might influence ZEN adsorption rate. The authors conclude that this specific clay has the potential to be used as a mycotoxin binder in poultry and probably for other animal species.

Unraveling the exact binding-mechanism, including the verification of the kinetics of mycotoxin binders has to this day remained largely uninvestigated. However, these data are of crucial importance to further ameliorate mycotoxin binders’ efficacies, and to aim for a wider adaption of these mitigation strategies in both animals and humans.

## 5. Materials and Methods

### 5.1. Chemical Products and Reagents

Methanol (LC-MS/MS grade) and glacial acetic acid (LC-MS/MS grade were procured from Biosolve B.V. (Valkenswaard, The Netherlands). Acetic acid and ammonium acetate (analytical grade) were supplied by Merck (Darmstadt, Germany). A Milli-Q^®^ SP Reagent water apparatus (Millipore; Brussels, Belgium) was used for water purification. For the mycotoxin standards, AFB1, AFB2, AFG1, AFG2 and ZEN were purchased from Sigma-Aldrich (Bornem, Belgium). Disinfectol^®^ (denaturated ethanol + 5% ether) was supplied by Chem-Lab (Belgium). Mycotoxin standards were dissolved in methanol (1 mg/mL), and were storable for a minimum of 1 year at −18 °C. Other chemicals and reagents were of analytical grade.

### 5.2. Source and Preparation of Clay

The raw clay sample was collected from Jebel Aïdoudi (El Hamma, Gabes) in the southern part of Tunisia. To purify the clay, the sample was ground and wet-sieved with an electromagnetic sieve shaker (Matest S.p.A., Triviolo, Italy) in order to eliminate all the impurities. Next, they were dried in an oven (Memmert GmbH + Co.KG, Schwabach, Germany) at 105 °C. The obtained clay was subsequently washed with distilled water until separation of the liquid/solid phases became difficult, and were dried again in an oven at 105 °C. The thermally-treated clay was obtained by a calcination process of the purified clay in a muffle furnace (Sirio Dentel Srl, Meldola, Italy) at 550 °C for 5h. Subsequently, purified clay (CP) and calcined clay (CC) samples were ground and sieved into fine powder (≈100 µm).

### 5.3. Physico-Chemical Characterization of Clays

#### 5.3.1. X-Ray Fluorescence (XRF)

The chemical composition of the minerals present in the two clays was performed using the XRF spectrometer model Thermo OASIS 9900 (Thermo Fisher Scientific (Schweiz) AG, Bâle, Suisse). It was performed in order to know the elemental oxides that are present in the clays.

#### 5.3.2. X-Ray Diffraction (XRD)

XRD was performed to identify the mineral composition of clays and the crystal line phases. It was conducted by an X-ray diffractometer « PANALYTICAL X’PERT PROMPD » using Cu Kα radiation (λ = 0.154 nm) with the voltage of 45 kV, 40 mA. Then the basal spacing of samples was determined from Bragg’s law (*n* λ= 2 d sin θ), where ‘n’ is the path difference between the reflected waves, equal to an integral number of wavelengths (λ), ‘λ’ the wavelength (nm), ‘d’ the interlayer spacing (nm), and ‘θ’ the diffraction angle (°).

#### 5.3.3. Fourier Transformed Infrared Spectroscopy (FTIR-ATR)

FTIR-IR helps in the identification of various forms of the minerals present in the clay. The vibrational spectra were performed with FTIR with a Spectrum Two PerkinElmer spectrometer, in the attenuated total reflection (ATR) mode, with a highly sensitive Deuterated Triglycine Sulfate (DTGS) detector. The samples were scanned 10 times in the range of 500–4000 cm^−1^, with a 2 cm^−1^ spectral resolution.

#### 5.3.4. Point of Zero Charge (PZC)

The Chemistry surface characterization was performed according to the solid addition method described by Noridine et al. [77] with slight modifications. A series of 50 mL of 0.01 mol/mL of NaCl (0.1 M) solutions were poured in beakers. The pH was adjusted in the range of 2–12 with 0.1 M HCl or 0.1 M NaOH solutions. Then, 0.2 g of clay was soaked in each solution under agitation at room temperature, and the final pH was measured after 24 h. The pH_PZC_ is the point in the curve when the pH_initial_ (pHi) verses pH_final_ (pHf) intersects the line (pHi = pHf).

#### 5.3.5. Cation Exchange Capacity (CEC)

CEC of clays was conducted using the barium chloride method [78]. Briefly, samples between 1 g and 5 g, depending on their probable CEC, were weighed into 50 mL centrifuge tubes. Then, 30 mL of 0.1 M BaCl_2_ was added, and the tubes were shaken slowly on a reciprocal shaker for 2 h. The samples were then centrifuged at 2500 rpm for 10 min and the supernatant solution were filtered through Whatman 42 filter paper. The solution was collected in polyethylene bottles for analysis.

#### 5.3.6. BET Surface Analysis

The textural properties of the clays (specific surface area, porosity) were determined by nitrogen adsorption using the multipoint Brunauer-Emmet-Teller (BET) method (ASAP 2020 Micromeritics Instruments, Norcross, GA, USA). The specific surface area was measured at 77 K and the pore size distribution was calculated in the radius range from 2000 to 100,000 nm by the BJH method using the adsorption isotherm.

### 5.4. Mycotoxin Adsorption

In vitro evaluation of the adsorption capacity of both clays for AFB1, AFB2, AFG1, AFG2, and ZEN was adopted from Di Mavungu et al. [79] with slight modifications. To determine the effect of pH on the mycotoxin binding capacity within the pH range of the gastrointestinal tract of poults, tests were performed at pH 3 and 7. In brief, 50 mg of clay sample was shaken and incubated for 3 h in 10 mL citrate buffer (pH 3) or phosphate buffer (pH 7) together with AFB1, AFB2, AFG1, and AFG2 at a concentration of 5 ng/mL, or with ZEN at a concentration of 25 ng/mL.

The mycotoxins were determined by LC-MS/MS using previously described methods [80]. A Waters Acquity UPLC system coupled to a Quattro Premier XE mass spectrometer (Waters, Milford, MA, USA) equipped with a Z-spray electrospray ionization (ESI) interface was used for the determination and quantification of mycotoxins. Chromatographic separation was achieved using a Symmetry C_18_ column (5 µm, 150 × 2.1 mm i.d.) with a Sentry guard column (3.5 µm, 10 × 2.1 mm i.d.) both supplied by Waters (Zellik, Belgium). The column was kept at room temperature. A mobile phase consisting of eluents A [water/methanol/acetic acid (94/5/1, *v*/*v*/*v*) containing 5 mM ammonium acetate] and B [methanol/water/ acetic acid (97/2/1, *v*/*v*/*v*) containing 5 mM ammonium acetate] was used at a flow rate of 0.3 mL min^−1^. A gradient elution was applied as follows: 0–7 min, 95% A/5% B –35% A/65% B; 7–11 min, 35% A/65% B–25% A/75% B; 11–13 min, 25% A/75% B–0% A/ 100% B; 13–15 min, 0% A/100% B; 15–16 min, 0% A/ 100% B–40% A/60% B; 16–22min, 40% A/60% B–60% A/40% B; 22–23 min, 60% A/40% B–95% A/5% B; 23–25 min 95% A/5% B. The injection volume was 20 µL.

### 5.5. Calculation of Mycotoxin Adsorption Rate (%)

The percentage of adsorption rate was calculated according to the following equation:% binding capacity= 100 × (Ci − Cf)/Ci(1)where

Ci is the initial concentration of mycotoxin

Cf is the concentration of unbound mycotoxin after incubation period

### 5.6. Calculation of the Binding Efficiency

The binding efficiency of the clays was determined according to the obtained results of the adsorption and desorption rate ((1) and (2))
% Desorption = ((% Adsorption pH3 − % Adsorption pH7)/% Adsorption pH3) × 100(2) with % binding efficiency = % Adsorption − % Desorption(3)

### 5.7. Statistical Analysis

Data were analyzed using the statistical software SPSS version 24 (IBM, New York, NY, USA) and results are presented as means ± SD. All analyses were performed in triplicate, and the significance of the means value was performed with the student’s t-test after determination of normality. Results were considered statistically significant at *p* ≤ 0.05.

## Figures and Tables

**Figure 1 toxins-11-00602-f001:**
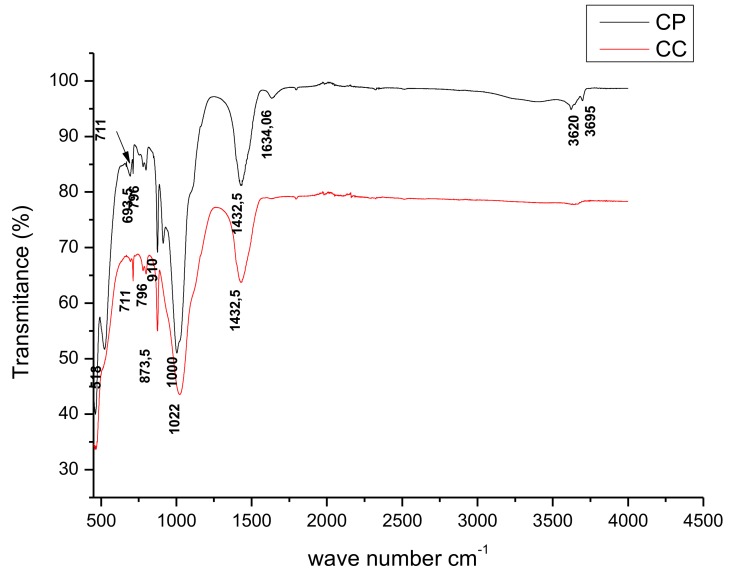
Infrared spectra of purified clay (CP) and calcinated clay (CC).

**Figure 2 toxins-11-00602-f002:**
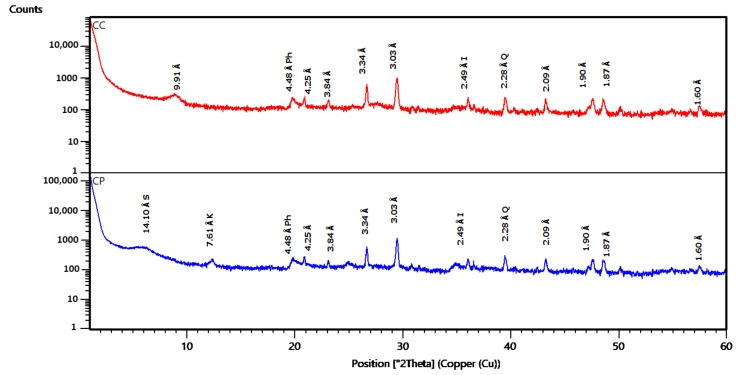
X-ray diffractograms of purified and calcined clay (S: smectite, I: illite, K: kaolinite, Q: quartz, C: calcite, D: dolomite), CP: purified clay, CC: calcined clay.

**Figure 3 toxins-11-00602-f003:**
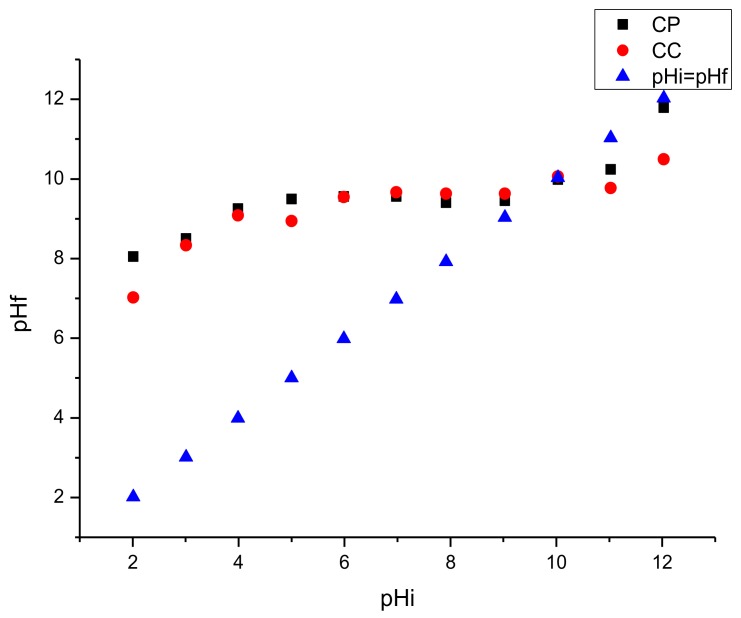
Determination of point of zero charge of purified clay (CP) and calcined clay (CC).

**Figure 4 toxins-11-00602-f004:**
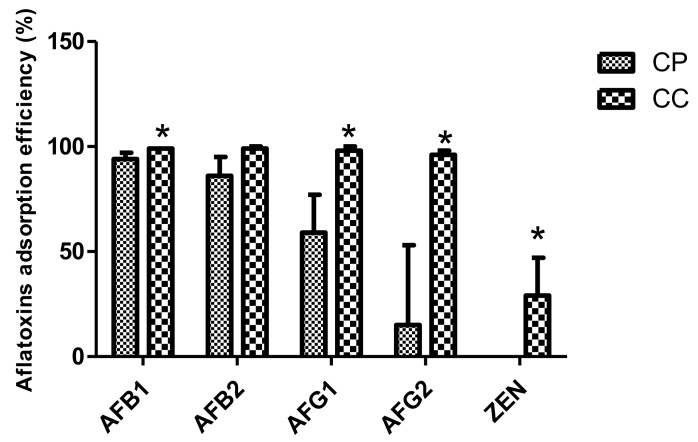
In vitro binding efficiency (%) of AFB1 (5 ng/mL), AFB2 (5 ng/mL), AFG1 (5 ng/mL), AFG2 (5 ng/mL) and ZEN (25 ng/mL) of purified clay (CP) and calcined clay (CC). Data are represented as mean values ± SD. Means of CC indicated with * are significantly different compared to CP (*p* < 0.05)

**Table 1 toxins-11-00602-t001:** Chemical composition of clays.

	Oxide Composition of the Clays (%)	
	CP	CC
SiO_2_	42.04	43.62
Al_2_O_3_	14.60	15.67
CaO	13.34	13.78
Fe_2_O_3_	11.03	9.69
K_2_O	1.12	1.17
MgO	1.74	1.78
Na_2_O	0.18	0.18
SO_3_	0.18	0.16

CP: purified clay, CC: calcined clay.

**Table 2 toxins-11-00602-t002:** Cation exchange capacity of purified clay (CP) and calcined clay (CC).

Clay Samples	Ca (mg/L)	K (mg/L)	Mg (mg/L)	Na (mg/L)	CEC (Cmol_(+)_(kg^−1^))
CP	126.18	14.1	24.44	17.94	12.266
CC	88.56	25.56	19	11.54	9.287

CP: purified clay, CC: calcined clay, CEC: cation exchange capacity.

**Table 3 toxins-11-00602-t003:** Textural characteristic of clay minerals.

Clay Samples	S_BET_ (m^2^/g)	Pore Volume (cm^3^/g)	Pore Size (Å)
CP	64.06	0.05	57.03
CC	44.42	0.05	66.72

S_BET_: BET Surface Area, CP: purified clay, CC: calcined clay.

**Table 4 toxins-11-00602-t004:** In vitro adsorption of AFB1, AFB2, AFG1, AFG2 and ZEN by purified clay (CP) and calcined clay (CC) at pH 3. Results are presented as mean ± SD. Means of CC indicated with * are significantly different compared to CP (*p < 0.05*). (NS: not significantly different).

Binding Capacity (%)			
Mycotoxins	CP	CC	*p-*Value
AFB1	100 ± 0	100 ± 0	*NS*
AFB2	88 ± 1	100 ± 0 *	*<0.001*
AFG1	96 ± 1	100 ± 0 *	*<0.001*
AFG2	76 ± 2	99 ± 1 *	*<0.001*
ZEN	0±0	75 ± 3 *	*<0.001*

**Table 5 toxins-11-00602-t005:** In vitro adsorption of AFB1, AFB2, AFG1, AFG2 and ZEN by purified clay (CP) and calcined clay (CC) at pH 7. Results are presented as mean ± SD. Means of CC indicated with * are significantly different compared to CP (*p* < 0.05).

Binding Capacity (%)			
Mycotoxins	CP	CC	*p-*Value
AFB1	94 ± 3	99 ± 0 *	*0.031*
AFB2	86 ± 8	99 ± 1 *	*0.048*
AFG1	60 ± 17	98 ± 2 *	*0.019*
AFG2	30 ± 29	96 ± 2 *	*0.017*
ZEN	1 ± 1	41 ± 12 *	*0.026*

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
