# Peer review of "Calcination Enhances the Aflatoxin and Zearalenone Binding Efficiency of a Tunisian Clay"

_toxins, 2019, doi:10.3390/toxins11100602_

Round 1

Reviewer 1 Report

The work titled as "Calcination enhances the aflatoxin and zearalenone binding efficiency of a Tunisian clay" is well structured and written, and the results clearly presented. In my opinion the work can be published after minor revision.

Equuation 1 for calculation of mycotoxin adsorption rate has to be checked. it looks like a typographic error.

Reference have to be checked. For instance,  there is a mistake in reference 21 in number and pages.

Author Response

Response to Reviewer 1 Comments

Point 1: Equation 1 for calculation of mycotoxin adsorption rate has to be checked. It looks like a typographic error.

Response 1: % binding capacity= 100 – (100 / Ci × Cf) is corrected to: % binding capacity= 100 x (Ci-Cf)/Ci

Point 2: Reference have to be checked. For instance, there is a mistake in reference 21 in number and pages.

Response 2: The list of the references was adopted according to the reviewer suggestion in the manuscript.

Reviewer 2 Report

The publication concerns a very important subject for mycotoxins adsorption by clays. Aflatoxins adsorption is already known and studied phenomenal, while new in relation to ZEN.

The aim of the study was to assess the effect of clay heat treatment on adsorption properties.

Regarding the results obtained, especially regarding aflatoxins, I have doubts whether the use of an energy-consuming method (550oC, 5 hours) is justified.

In addition, the authors placed great emphasis on studying the physicochemical properties of clays. Toxin adsorption studies are just two short experiments. The publication lacks an attempt to clearly combine these two elements and define the relationship between them.

In my opinion, the article can be published in toxins but after major revision. Below are my specific comments.

Introduction

L.34 – lack of dot and space after [3,4]

L.41 – should be [8-10]

L.57 – should be [8,23]

L49-64 - in general, the authors did not address the disadvantages of using clay as adsorbents, i.e. reducing the nutritional value of decontaminated feed.

Results

In my opinion parts 2.1-2.6 are too extensive. There is no statistical analysis of the results.

It seems to me that the authors should not use the phrase “desorption” (Tab. 5) - in my opinion, this is adsorption at a different pH than in Table 4.

In addition, the results presented in Figure 4 are from the same experiment.

The use of two pH values 3 and 7 as the GI tract model is a significant simplification.

Discussion

This part is written quite well however in my opinion this part should be more developed in the field of mycotoxin adsorption mechanisms by clays with different adsorption capacity.

Methods Please explain in detail the use of the term "desorption". I have doubts as to whether it is appropriate as presented by the authors

References

The list of literature needs to be corrected and supplemented. Many items are incomplete, eg. journal name missing: 5, 7, 8, 11, 12, 18, 26, 27, 28, 30, 32, 49, 50, 51, 53, 56, 57, 58, 59, 60. This is unacceptable.

Author Response

Point 1: Regarding the results obtained, especially regarding aflatoxins, I have doubts whether the use of an energy-consuming method (550°C, 5hours) is justified. 

Response 1: Calcination of clay minerals is used to improve the raw clay characteristics (Ediz et al., 2012). Calcined clay at different temperature have been used in numerous studies as adsorbents to eliminate heavy metals, cationic dyes, pesticides and other environmental toxins (Chen et al., 2011, Vieira et al., 2010, Bojemueller et al., 2001, Tlili et al., 2012). Chaari et al., (2008), reported that the removal of lead by calcined clay increase with increasing temperature. In addition, Bojemueller et al., (2001), reported that the value of calcined products in the adsorption depends on the calcination temperature and also on the period of heating. He reported also that a period of 3h and a temperature of 550°C reduced the specific surface area and the porosity, while with increasing temperature, the surface area and the porosity increase which lead to better adsorption.

To the best of our knowledge, only one study investigated the effect of calcined clay (250 °C for 1h) on mycotoxin binding (Nones et al., 2015), in this paper it was reported that the adsorption of AFB1 was reduced after calcination of a bentonite clay. Herein, the novelty of our study is to investigate the effect of calcined clay (at 550° C, 5h) in reducing the toxic effect of mycotoxin. Furthermore, there is no heat involved in our experiment with mycotoxins, the heat treatment of clay was done before doing the experiment.

Ediz, N., Bentli, İ., Tatar, İ. Improvement in filtration characteristics of diatomite by calcination. 2010 International Journal of Mineral Processing 94, 129–134. Chen, H.; Zhao, J.; Zhong, A.; Jin, Y. Removal capacity and adsorption mechanism of heat-treated palygorskite clay for methylene blue. Chemical Engineering Journal 2011, 174, 143–150. Vieira, M.G.A.; Neto, A.F.A.; Gimenes, M.L.; da Silva, M.G.C. Removal of nickel on Bofe bentonite calcined clay in porous bed. Journal of Hazardous Materials 2010, 176, 109–118. Bojemueller, E.; Nennemann, A.; Lagaly, G. Enhanced pesticide adsorption by thermally modified bentonites. Applied Clay Science 2001, 18, 277–284. Tlili, A.; Saidi, R.; Fourati, A.; Ammar, N.; Jamoussi, F. Applied Clay Science Mineralogical study and properties of natural and fl ux calcined porcelanite from Gafsa-Metlaoui basin compared to diatomaceous fi ltration aids. Applied Clay Science 2012, 6263, 47–57. Chaari, I.; Fakhfakh, E.; Chakroun, S.; Bouzid, J.; Boujelben, N. Lead removal from aqueous solutions by a Tunisian smectitic clay. Journal of Hazardous Materials 2008, 156, 545–551. Nones, J.; Nones, J.; Gracher, H.; Poli, A.; Gonçalves, A.; Cabral, N. Thermal treatment of bentonite reduces aflatoxin B1 adsorption and affects stem cell death ☆. Materials Science & Engineering C 2015, 55, 530–537.

Point 2: In addition, the authors placed great emphasis on studying the physico-chemical properties of clays. Toxin adsorption studies are just two short experiments. The publication lacks an attempt to clearly combine these two elements and define the relationship between them.

Response 2: The physico-chemical characterization of the purified and calcinated clay has been strongly emphasized because we want to now the characteristic of the studied clays which are never been used before, also, it allows us to conclude the change of their characteristic before and after calcination. Furthermore, it is documented that the change in characteristic of clay is important for binding mycotoxins (None et al., 2015, Deng et al., 2012).

Nones, J.; Nones, J.; Gracher, H.; Poli, A.; Gonçalves, A.; Cabral, N. Thermal treatment of bentonite reduces aflatoxin B1 adsorption and affects stem cell death ☆. Materials Science & Engineering C 2015, 55, 530–537. Deng, Y., Liu, L., Barrientos Velázquez, A.L., Dixon, J.B. The determinative role of the exchange cation and layer-charge density of smectite on aflatoxin adsorption. Clay Clay Mineral 2012, 60, 374–386.

Introduction

Point 3: L.34 – lack of dot and space after [3,4]

Response 3: It was corrected in the manuscript. Please see L.35

Point 4: L.41 – should be [8-10]

Response 4: ([8-10]) is corrected to [8-10]. Please see L.42

Point 5:  L.57 – should be [8,23]

 Response 5: ([8,23]) is corrected to [8,23]. Please see L.58

Point 6: L49-64 – in general, the authors did not address the disadvantages of using clay as adsorbents, i.e. reducing the nutritional value of decontaminated feed.

Response 6: Disadvantages of using clay as adsorbents were added. Please see L58-61.

Results

Point 7: In my opinion parts 2.1 – 2.6 are too extensive. There is no statistical analysis of the results

Response 7:

Taking into account the reviewer suggestion, we reduced the parts 2.1 – 2.6. in the manuscript. Please see L. 85-131

Point 8: It seems to me that the authors should not use the phrase “desorption” (Tab.5) – in my opinion, this is adsorption at a different pH than in table 4.

Response 8: In our study we calculate the desorption rate of mycotoxins based on the concentration of the adsorption rate.

Point 9: In addition, the results presented in Figure 4 are from the same experiment.

Response 9: Based on the in vitro adsorption data, the in vitro binding efficiency of AFs and ZEN of the purified and calcined clay was predicted. According to the reviewer comment we did some modifications to the results in the manuscript. Please see L.134-161.

Point 10:  The use of two pH values 3 and 7 as the GI tract model is a significant simplification.

Response 10: The use of pH values 3 and 7 as the GI model were successfully applied in previous in vitro experiments (Prapapanpong et al., 2019, Ayo et al., 2018, Daković et al., 2005).

Prapapanpong, J.; Udomkusonsri, P.; Mahavorasirikul, W.; Choochuay, S.; Tansakul, N. In Vitro Studies on Gastrointestinal Monogastric and Avian Models to Evaluate the Binding Efficacy of Mycotoxin adsorbents by Liquid Chromatography-Tandem Mass Spectrometry. Journal of Advanced Veterinary and Animal Research 2019, 6, 125–132. Ayo, E.M.; Matemu, A.; Laswai, G.H.; Kimanya, M.E. An In Vitro Evaluation of the Capacity of Local Tanzanian Crude Clay and Ash-Based Materials in Binding Aflatoxins in Solution. Toxins 2018, 1, 1–13. Daković, A.; Kragović, M.; Rottinghaus, G.E.; Sekulić, Ž.; Milićević, S.; Milonjić, S.K.; Zarić, S. Influence of natural zeolitic tuff and organozeolites surface charge on sorption of ionizable fumonisin B1. Colloids and Surfaces B: Biointerfaces 2010, 76, 272–278.

Discussion

Point 11: This part is written quite well however in my opinion this part should be more developed in the field of mycotoxin adsorption mechanisms by clays with different adsorption capacity.

Response 11: According to the reviewer suggestion we have developed the discussion part in the manuscript.

Methods

Point 12: Please explain in detail the use of the term “desorption”. I have doubts as to whether it is appropriate as presented by the authors.

Response 12:

In our study we calculate the desorption rate of mycotoxins based on the concentration of the adsorption rate.

Calculation of mycotoxin desorption rate (%)

% Desorption= ((% Adsorption pH3 - % Adsorption pH7) /pH3) × 100

References

Point 13: The list of literature needs to be corrected and supplemented many items are incomplete, eg journal name missing: 5, 7, 8, 11, 12, 18, 26, 27, 28, 30, 32, 49, 50, 51, 53, 56, 57, 58, 59, 60. This is unacceptable.

Response 13:

The list of the references was adopted according to the reviewer suggestion in the manuscript.

Round 2

Reviewer 2 Report

Paper in its current form is suitable for publication in Toxins.

The authors responded to all the comments from reviewer.